# Salicylic Acid and Pyraclostrobin Can Mitigate Salinity Stress and Improve Anti-Oxidative Enzyme Activities, Photosynthesis, and Soybean Production under Saline–Alkali Regions

**Honglei Ren** [1,†]**, Xueyang Wang** [1,†]**, Fengyi Zhang** [1]**, Kezhen Zhao** [1]**, Xiulin Liu** [1]**, Rongqiang Yuan** [1]**, Changjun Zhou** [2]**, Jidong Yu** [2]**, Jidao Du** [3]**, Bixian Zhang** [1,*] **and Jiajun Wang** [1,*]

[1] Soybean Research Institute, Northeastern Precocious Soybean Scientific Observation Station of Ministry of Agriculture and Rural Affairs, Heilongjiang Academy of Agriculture Sciences, Harbin Branch of National Soybean Improvement Center, Harbin 150086, China; midou@haas.cn (H.R.); hljsnkywxy@163.com (X.W.); ddszhangfy2019@163.com (F.Z.); zhaokz928@163.com (K.Z.); liuxiulin1002@126.com (X.L.); yrq18846121189@163.com (R.Y.)

[2] Daqing Branch of Heilongjiang Academy of Agricultural Sciences, Daqing 163316, China; andazhouchangjun@163.com (C.Z.); yujidong666@126.com (J.Y.)

[3] Key Laboratory of Soybean Mechanized Production, Ministry of Agriculture and Rural Affairs, College of Agriculture, Heilongjiang Bayi Agricultural University, Daqing 163319, China; djdbynd@163.com

[*] Correspondence: hljsnkyzbx@163.com (B.Z.); junjiawang@163.com (J.W.)

[†] These authors contributed equally to this work.

**Abstract:** Soybean is a widespread crop in semi-arid regions of China, where soil salinity often increases and has a significant harmful impact on production, which will be a huge challenge in the coming years. Salicylic acid (SA) and pyraclostrobin are strobilurin-based bactericides (PBF). Under rainfall-harvesting conditions in covered ridges, the exogenous application of SA and PBF can improve the growth performance of soybeans, thereby reducing the adverse effects of soil salinity. The objectives of this research are to evaluate the potential effects of SA and PBF on soybean growth in two different regions, Harbin and Daqing. A two-year study was performed with the following four treatments: HCK: Harbin location with control; SA1+PBF1: salicylic acid (5 mL L$^{-1}$) with pyraclostrobin (3 mL L$^{-1}$); SA2+PBF2: salicylic acid (10 mL L$^{-1}$) with pyraclostrobin (6 mL L$^{-1}$); DCK: Daqing location with control. The results showed that in the Harbin region, SA2+PBF2 treatment reduced the evapotranspiration (ET) rate, increased soil water storage (SWS) during branching and flowering stages, and achieved a maximum photosynthesis rate. Moreover, this improvement is due to the reduction of MDA and oxidative damage in soybean at various growth stages. At different growth stages, the treatment of Harbin soybean with SA2+PBF2 significantly increased the activity of CAT, POD, SOD, and SP, while the content of MDA, $H_2O_2$, and $O_2^-$ also decreased significantly. In the treatment of SA2+PBF2 in Harbin, the scavenging ability of free $H_2O_2$ and $O_2^-$ was higher, and the activity of antioxidant enzymes was better. This was due to a worse level of lipid-peroxidation which successfully protected the photosynthesis mechanism and considerably increased water use efficiency (WUE) (46.3%) and grain yield (57.5%). Therefore, using plastic mulch with SA2+PBF2 treatment can be an effective water-saving management strategy, improving anti-oxidant enzyme activities, photosynthesis, and soybean production.

**Keywords:** soil salinity; salicylic acid; anti-oxidant metabolism; photosynthesis; reactive oxygen species; soybean production



## 1. Introduction

High salinity is one of the key non-biotic stresses that leads to the decline of agricultural production [1]. Given the cultivation of soybeans in the dry-land farming systems, this issue is particularly important [2]. By 2050, 50% of farming land may be affected by salinity [3]. Especially, the threat of salinization has an impact on the soil in coastal agricultural areas [4].

However, the increase in the salinity of arable land may affect the sustainable food supply of the world population [5]. Soybean is the world's most essential oilseed crop and a significant component of worldwide food security [6]. The shortage of water resources, high soil salinity, evaporation rate, and low soil fertility status has severely restricted the improvement of the dryland farming systems [7]. A ridge and furrow rainfall-harvesting system with plastic mulching can effectively raise SWS and reduce salt content and is widely used to promote soybean production and WUE [8,9]. Plastic mulching is also appropriate for improving the photosynthesis and yield of soybean in the saline–alkali regions [10].

Salinity is harmful to soybean because it causes negative physiological, biochemical, and morphological effects and can lead to a decrease in crop production [3]. The plant growth reduction caused by salinity is mostly determined by the following aspects: (i) the increase of osmotic pressure of the culture medium reduces the ability of plants to absorb water [11]; (ii) the toxicity level of excessive ions on plant cells [12]; and (iii) ionic imbalances that affect plant nutritional status and affect biochemical and metabolic components linked to planting development [13]. Plants have developed an inherent defense mechanism, including limitations on the absorption of toxic ions, stomatal regulation to sustain water status under salt stress, and enzymatic and non-enzymatic antioxidants [14]. Salt stress and other environmental stresses can raise the production of ROS, leading to oxidative stress in plant cells [15]. ROS has high cytotoxicity and can react with different biological molecules, leading to DNA mutations, lipid peroxidation, protein denaturation, and death of cell [16]. ROS elimination can be attained by activating the antioxidant mechanism [17]. Although defense mechanisms may be compromised by severe salinity stress, plants may be adversely affected by salinity-induced damage.

We use some strategies for using biostimulants that can improve the biochemical processes of plants under stress [18]. Due to the relationship in many aspects of plant physiological responses to salt and water stress, it is assumed that pyraclostrobin-based fungicides (PBF) can reduce the destructive effects of salinity [19]. These natural molecules are produced by a group of fungi belonging to the phylum basidiomycota, which is a pathogen of wood decay in certain tree species [20]. The bactericidal effect is achieved by blocking the outer electron transport to inhibit the mitochondrial respiration of some agricultural crop pathogenic fungi [21]. In recent years, several studies have proven that crop protection products based on astragalus have complementary biological stimulator properties, which can reduce water demand and alleviate non-biotic stress [22]. In fact, after the application of these pyraclostrobin-based fungicides (PBF), changes in plant metabolism have been observed, including an increase in ABA production and the activation of oxidative stress enzymes [23], which may increase WUE and photosynthesis under salt stress [24,25].

Many growth regulators, including hormones that have been used to activate plants, seem to be promising technologies for reducing salt toxicity and improving plant yield [16,26]. SA is an important phenolic compound and a growth regulator that plays a unique role in plants' various biochemical processes [27,28]. SA can also reduce lipid peroxidation and improve plant resistance to salt stress [29]. Drought and salt stress can promote the production of ROS, such as $H_2O_2$ and $O_2^-$, leading to chlorophyll damage [30]. Due to the increased accumulation of ROS, plant water and leaf stresses are typically linked with improved activation of oxidative stress enzymes [31]. Oxidative damage can have a negative impact on Pn and chlorophyll content [32,33]. Therefore, an approach to protect soybean from oxidative damage and delay the aging process is crucially important for its anti-oxidant metabolism [34,35].

Due to the commercial significance of soybean crops, the spread of salinity issues in several regions covered by these crops, and economic losses, there is increasing interest in finding new solutions that can alleviate the negative impact of salinity. We assume that PBF may not only have bactericidal effects but also biologically stimulating effects, which can decrease the harmful effects of salt stress on soybean. Our research objectives are as follows: (i) to examine the interaction between PBF and SA strategies in the antioxidant defense system of soybean leaves and the improvement of water use efficiency; (ii) to

determine the changes in the photosynthetic capacity and ROS detoxification system of soybean leaves under plastic film mulching.

## 2. Materials and Methods

### 2.1. Site Description

This study was conducted in Daqing, located at (125°19′16.59″ E, 46°62′5.31″ N) and 147.5 m asl, and the second site was in Harbin, located at (126°51′41.91″ E, 45°50′37.82″ N) and 174 m asl, China. The total sunshine hours were 2345.2 h yr$^{-1}$. Over 60% of the rainfall occurred from June to September. Rainfall during the soybean-growing season in Harbin from 2020 to 2021 was 398 mm and 351 mm, while Daqing had 510 mm and 470 mm (Figure 1). The initial physicochemical properties of the soil were determined; the soil samples were collected at a depth of 20 cm at the two experimental sites, which is shown in Table 1.

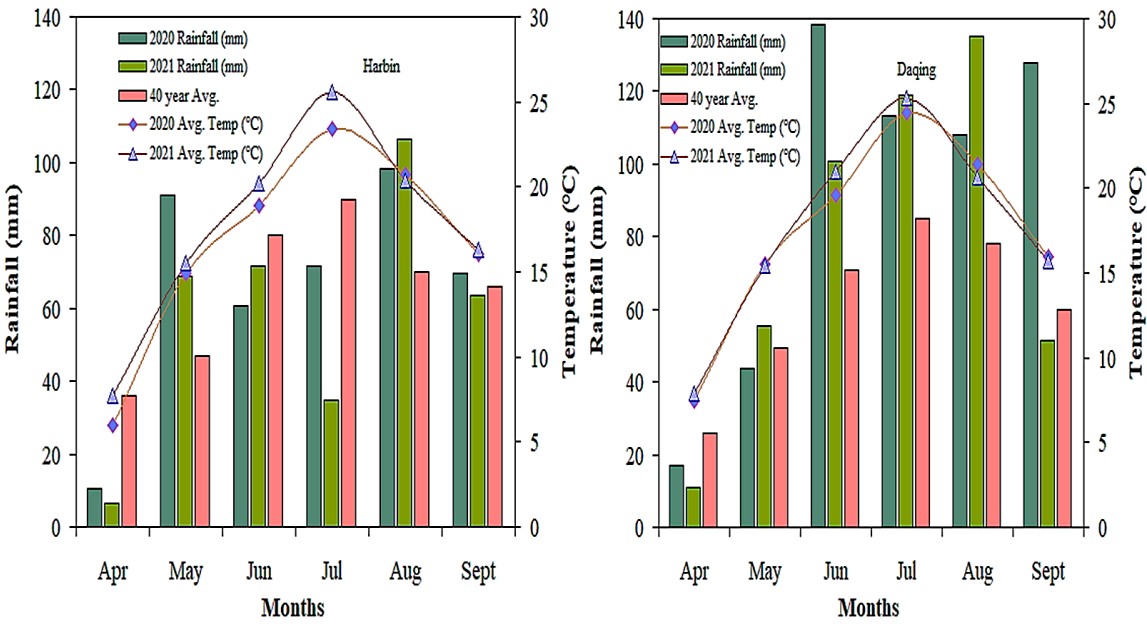

**Figure 1.** Precipitation distribution during 2020 and 2021 soybean-growing seasons.

**Table 1.** The chemical properties of experimental site of the soil layers (0–20 cm).

| Location | SOM (g kg$^{-1}$) | TN (g kg$^{-1}$) | TP (g kg$^{-1}$) | TK (g kg$^{-1}$) | AN (mg kg$^{-1}$) | AP (mg kg$^{-1}$) | AK (mg kg$^{-1}$) |
|---|---|---|---|---|---|---|---|
| Harbin | 32.20 | 0.17 | 1.07 | 2.71 | 167.05 | 24.81 | 162.22 |
| Daqing | 29.89 | 0.18 | 1.05 | 2.48 | 135.81 | 23.95 | 116.70 |

### 2.2. Research Design

The field research was conducted in a randomized complete block design with three replications. In 2020 and 2021, we conducted field research under two experimental locations in Harbin and Daqing to study the effects of four different treatments: HCK: Harbin location with control; SA1+PBF1: salicylic acid (5 mL L$^{-1}$) with pyraclostrobin (3 mL L$^{-1}$); and SA2+PBF2: salicylic acid (10 mL L$^{-1}$) with pyraclostrobin (6 mL L$^{-1}$). Meanwhile, the Daqing location had DCK: Daqing location with control; SA1+PBF1: salicylic acid (5 mL L$^{-1}$) with pyraclostrobin (3 mL L$^{-1}$); and SA2+PBF2: salicylic acid (10 mL L$^{-1}$) with pyraclostrobin (6 mL L$^{-1}$).. Each ridge and furrow was 65 cm wide and 15 cm high. Soybeans were planted along ridges and furrows. A 1-meter wide isolation strip was set up between each plot to prevent water leakage. Heinong-531 soybean variety was planted on 17 May 2020 and harvested on 10 October 2020; the corresponding dates are 12 May and 15 October 2021. During the entire growing season, field management was carried out in

accordance with local practices. The amount of fertilizer applied during soil preparation each year was 60, 90, and 75 kg N, $P_2O_5$, $K_2O$, $ha^{-1}$.

### 2.3. Soil Water Storage (SWS)

The soil water content was calculated at the seedling, branching, flowering, grain-filling, and maturity stages during 2020 and 2021. Moisture contents of the 0–200 cm soil layers at 20 cm intervals were recorded using a TDR meter (Time-Domain Reflectometry, Germany).

The SWS was calculated by the following formula [36]:

$$SWS = C \times \rho \times H \times 10 \tag{1}$$

where C is the soil water, $\rho$ is the bulk density, and H is the soil depth.

The ET rate was calculated by the following formula [36]:

$$ET = R + \Delta SWS \tag{2}$$

where R is the rainfall and $\Delta SWS$ is the SWS at a depth of (0–200 cm) between sowing and harvesting.

$$WUE = Y/ET \tag{3}$$

where WUE is the water use efficiency and Y is the grain yield.

### 2.4. Net Photosynthetic Rate (Pn)

Pn was measured using the LI Cor LI-6400XT portable photosynthesis system (LI-6400XT, LI Cor, Lincoln, NE, USA). Fully inflated leaves were measured on sunny days from 9:00 to 11:00 a.m. The $CO_2$ concentration in the leaf chamber was set to 380 $\mu$ mol $mol^{-1}$, and the photosynthetic active radiation was set to 1100 $\mu$ mol $m^{-2}\,s^{-1}$. Nine leaves of five individual plants were analyzed during the flowering, filling, and maturity stages of each treatment, with three replicates.

### 2.5. Antioxidant Enzyme Activities

An amount of 0.5 g of soybean leaf homogenate with midrib was removed, with 5 mL 50 mmol $L^{-1}$ phosphate buffer (pH 7.8), 0.1mM EDTA-$Na^2$, and 1% insoluble PVP. The homogenate was centrifuged at 15,000$\times$ *g* for 10 min at 40 °C. After centrifugation, the upper supernatant was taken and used for enzyme determination. The SOD activity was analyzed at 560 nm according to the technology of [37]. According to Amalo et al. [38], the POD activity was calculated using guaiacol at 470 nm. The determination of CAT activity was based on the method proposed by Tan et al. [39]. The MDA content was analyzed based on the technique proposed by Zhang [40]. The Coomassie Brilliant Blue (G250) method described by Read and Northcote [41] was used to measure the soluble protein contents. The $H_2O_2$ and $O_2^-$ contents were determined by the modification of the method of Elstner and Heupel [42] described by Jiang and Zhang [43]. The $O_2^-$ content was analyzed at 530 nm, while $H_2O_2$ content was analyzed at 415 nm [44].

### 2.6. Statistical Analysis

The data were analyzed using SPSS 18.0 software. Multiple comparisons were tested using Duncan's new multiple-range test. If the F-test was significant, the mean value was evaluated by a multiple comparison test (LSD 0.05).

### 3. Results

### 3.1. Soil Water Storage (SWS) and ET Rate

Typically, significant changes in soil temperature, ET, and rainwater utilization can lead to significant differences in SWS (0–200 cm) at various growth stages (Table 2). During the two years of the seedling stage, the SWS was roughly the same without significant differences. In a two-year study, SWS varied with precipitation and different growth stages.

From the SS to BS stages in Harbin and Daqing, the average SWS (0–2 m) of SA1+PBF1 and SA2+PBF2 treatments significantly increased by 14.8%, 20.4%, 8.4%, and 5.6% compared to HCK and DCK treatments, respectively. From FS to the GFS stage, SWS gradually increases. From the GFS stage to the MS stage, the SWS trend of each treatment improved compared to the FS stage. During the GFS to MS periods, the average SWS (0–200 cm) of SA1+PBF1 and SA2+PBF2 treatments in the Harbin region significantly increased by 22.2% and 34.3% compared to HCK treatment, while the MS SWS of SA1+PBS1 and SA2+PBS2 treatments in the Daqing region considerably improved by 1.8% and 6.4% compared with DCK.

**Table 2.** Soil water storage (mm) at 0–200 cm soil profile at various growth stages of soybean as affected by various treatments. [a] during 2020 and 2021.

| Locations | Treatments | Soil Water Storage (mm) | | | | | | | | | |
|---|---|---|---|---|---|---|---|---|---|---|---|
| | | **2020** | | | | | **2021** | | | | |
| | | **SS** | **BS** | **FS** | **GFS** | **MS** | **SS** | **BS** | **FS** | **GFS** | **MS** |
| Harbin | HCK | 118.5 a | 123.6 b | 97.7 c | 101.1 d | 92.0 d | 114.4 a | 129.7 b | 89.9 c | 84.9 e | 122.7 c |
| | SA1+PBF1 | 119.4 a | 126.1 a | 104.0 b | 113.4 b | 116.8 b | 115.5 a | 132.1 a | 99.8 b | 101.2 b | 129.7 b |
| | SA2+PBF2 | 119.9 a | 127.3 a | 115.3 a | 125.7 a | 129.7 a | 116.1 a | 133.2 a | 110.7 a | 111.7 a | 141.2 a |
| Daqing | DCK | 118.7 a | 124.3 b | 102.2 b | 107.2 c | 107.2 c | 114.6 a | 130.4 a | 80.5 d | 73.5 f | 110.1 d |
| | SA1+PBF1 | 119.5 a | 126.5 a | 101.6 b | 107.8 c | 110.5 c | 115.8 a | 132.1 a | 96.0 b | 92.0 d | 123.0 c |
| | SA2+PBF2 | 120.7 a | 126.0 a | 103.2 b | 112.7 b | 115.6 b | 117.1 a | 132.1 a | 99.4 b | 99.9 c | 128.4 b |
| Analysis of variance | L | * | * | * | * | * | * | * | * | * | * |
| | SA | * | * | * | * | * | * | * | * | * | * |
| | PBF | * | * | * | * | * | * | * | * | * | * |
| | L × SA | ns | ns | ns | ns | ns | ns | ns | ns | ns | ns |
| | L × PBF | ns | ns | ns | ns | ns | ns | ns | ns | ns | ns |
| | SA × PBF | * | * | * | * | * | * | * | * | * | * |
| | L × SA × PBF | ns | ns | ns | ns | ns | ns | ns | ns | ns | ns |

[a] HCK: Harbin location with control; SA1+PBF1: salicylic acid (5 mL L$^{-1}$) with pyraclostrobin (3 mL L$^{-1}$); SA2+PBF2: salicylic acid (10 mL L$^{-1}$) with pyraclostrobin (6 mL L$^{-1}$); DCK: Daqing location with control. Abbreviations are SS: seedling stage, BS: branching stage, FS: flowering stage, GFS: grain-filling stage, MS: maturity stage.

During the soybean growing season, there were significant variations in ET between different treatments. The soybean growth in the Daqing location had a high ET rate as compared with the Harbin location under different salicylic acid and pyraclostrobin levels. In 2020, the ET rate was significantly decreased at the location of Harbin with SA1+PBF1 and SA2+PBF2 treatments by 23.8% and 11.1% compared with HCK treatment, while at the Daqing location, SA1+PBF1 and SA2+PBF2 treatments had significantly decreased the ET rate by 22.4% and 9.0% compared with DCK treatment. In 2021, the ET rate was significantly decreased at the Harbin location with SA1+PBF1 and SA2+PBF2 treatments by 15.3% and 3.6% compared with HCK treatment, while at the Daqing location, SA1+PBF1 and SA2+PBF2 treatments had significantly decreased the ET rate by 16.4% and 11.3% compared with DCK treatment.

*3.2. Pn Rate and SP Content*

At various growth stages, the Pn and SP content of soybean leaves considerably improved with the increase of salicylic acid and pyraclostrobin levels. SP content is closely related to the Pn rate, significantly increasing the Pn rate of leaves. Due to the improvement of SWS, the SP content in soybean leaves was higher, which can maintain a higher net Pn rate (Table 3; Figure 2). During the two years of the same treatment, the Pn and SP contents of soybean leaves were considerably higher from the BS to FS stages, while they decreased considerably from the FS to GFS stages (Table 3; Figure 2). The average Pn was significantly improved at the Harbin location with SA1+PBF1 and SA2+PBF2 treatments by 36.9% and 47.9% compared with HCK treatment, while at the Daqing location, SA1+PBF1 and SA2+PBF2 treatments had 46.4% and 54.4% higher Pn compared with DCK. In a two-

year study, the Pn and SP content of SA1+PBF1 and SA2+PBF2 was considerably higher than those of HCK and DCK in the BS, FS, and GFS stages. However, with the treatment of SA1+PBF1, there was no considerable difference in Pn and SP content between the two study sites at various growth stages. In the later stage of soybean growth, both Pn and SP content was significantly affected by the levels of salicylic acid and pyraclostrobin, unlike in HCK and DCK treatments.

**Table 3.** The net photosynthesis rate of soybean as affected by various treatments during 2020 and 2021.

| Locations | Treatments | Net Photosynthesis Rate (Pn µmol. m$^{-2}$ s$^{-1}$) | | | | | |
| | | 2020 | | | 2021 | | |
| | | BS | FS | GFS | BS | FS | GFS |
|---|---|---|---|---|---|---|---|
| Harbin | CK | 16.9 d | 17.8 c | 7.2 d | 13.9 d | 19.6 c | 8.9 d |
| | SA1+PBF1 | 23.2 b | 28.3 b | 13.2 b | 19.9 b | 25.8 b | 11.6 c |
| | SA2+PBF2 | 26.2 a | 32.0 a | 17.0 a | 25.0 a | 28.6 a | 19.0 a |
| Daqing | CK | 11.9 e | 15.1 d | 5.7 e | 12.3 d | 14.6 d | 6.5 e |
| | SA1+PBF1 | 21.6 c | 27.0 b | 9.1 c | 16.8 c | 24.4 b | 9.6 d |
| | SA2+PBF2 | 24.2 b | 28.6 b | 12.9 b | 20.8 b | 25.6 b | 15.8 b |
| | L | * | * | * | * | * | * |
| | SA | * | * | * | * | * | * |
| | PBF | * | * | * | * | * | * |
| Analysis of variance | L × SA | ns | ns | ns | ns | ns | ns |
| | L × PBF | ns | ns | ns | ns | ns | ns |
| | SA × PBF | * | * | * | * | * | * |
| | L × SA × PBF | ns | ns | ns | ns | ns | ns |

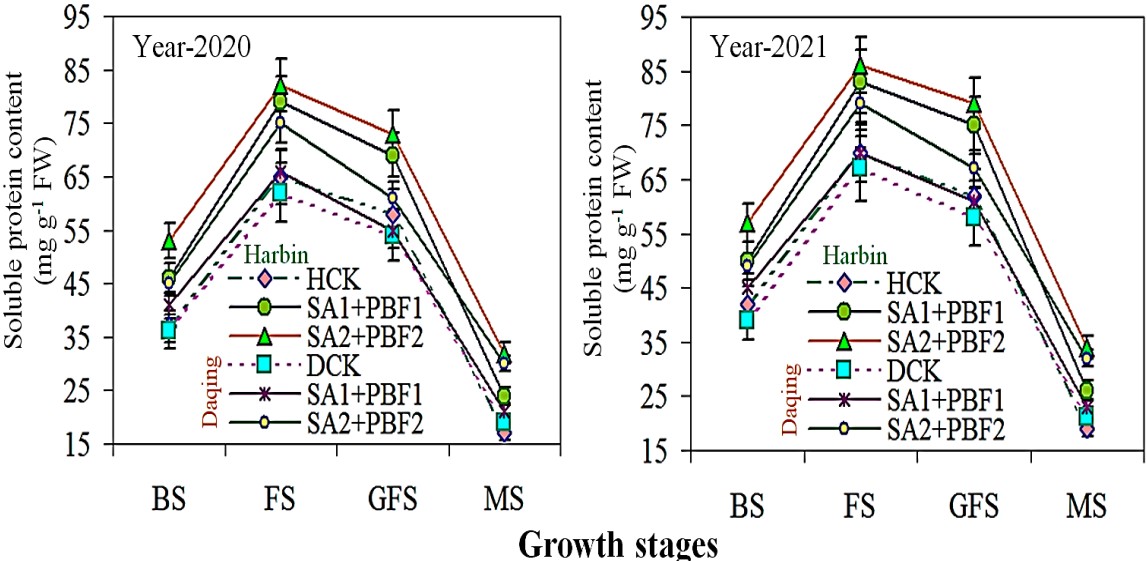

**Figure 2.** Effects of different treatments on soluble protein content (SP) of soybean during 2020–2021 growing seasons. Note: HCK: Harbin location with control; SA1+PBF1: salicylic acid (5 mL L$^{-1}$) with pyraclostrobin (3 mL L$^{-1}$); SA2+PBF2: salicylic acid (10 mL L$^{-1}$) with pyraclostrobin (6 mL L$^{-1}$); DCK: Daqing location with control.

### 3.3. Anti-Oxidative Enzyme Activities

Tables 4 and 5 and Figure 3 show that the POD, CAT, and SOD activities of soybean leaves considerably improved with increasing levels of salicylic acid and pyraclostrobin during the BS, FS, GFS, and MS stages at the research sites in Harbin and Daqing. The SA2+PBF2 treatment had the highest POD, CAT, and SOD activities in soybean leaves during the FS stage, but there was no significant difference compared to the GFS stage. Afterward, the SOD, POD, and CAT activities in soybean leaves rapidly decreased during

the MS stage. Furthermore, there were no considerable variations between the SA1+PBF1 treatments at all growth stages of soybeans at the research sites in Harbin and Daqing.

**Table 4.** Peroxidase (POD) activity of soybean as affected by various treatments during 2020 and 2021.

| Locations | Treatments | POD Activity (U g$^{-1}$ FW min$^{-1}$) | | | | | | | |
|---|---|---|---|---|---|---|---|---|---|
| | | 2020 | | | | 2021 | | | |
| | | **BS** | **FS** | **GFS** | **MS** | **BS** | **FS** | **GFS** | **MS** |
| Harbin | CK | 93.8 e | 191.2 d | 135.1 e | 110.1 e | 96.8 e | 197.2 d | 141.1 e | 113.0 e |
| | SA1+PBF1 | 116.0 c | 314.0 b | 312.8 b | 159.4 c | 120.0 b | 326.0 b | 319.4 b | 166.3 c |
| | SA2+PBF2 | 140.5 a | 359.7 a | 330.1 a | 240.0 a | 144.5 a | 349.7 a | 336.7 a | 246.9 a |
| Daqing | CK | 93.9 e | 135.5 e | 127.0 f | 117.1 d | 99.9 d | 145.5 e | 133.9 f | 123.3 d |
| | SA1+PBF1 | 96.9 d | 191.7 d | 194.1 d | 157.1 c | 100.9 c | 203.7 d | 200.7 d | 164.0 c |
| | SA2+PBF2 | 136.3 b | 237.1 c | 228.6 c | 222.6 b | 140.3 a | 271.1 c | 235.2 c | 229.5 b |
| | L | * | * | * | * | * | * | * | * |
| | SA | * | * | * | * | * | * | * | * |
| Analysis | PBF | * | * | * | * | * | * | * | * |
| of | L × SA | ns | ns | ns | ns | ns | ns | ns | ns |
| variance | L × PBF | ns | ns | ns | ns | ns | ns | ns | ns |
| | SA × PBF | * | * | * | * | * | * | * | * |
| | L × SA × PBF | ns | ns | ns | ns | ns | ns | ns | ns |

**Table 5.** Catalase (CAT) activity of soybean as affected by various treatments during 2020 and 2021.

| Locations | Treatments | CAT Activity (U g$^{-1}$ FW min$^{-1}$) | | | | | | | |
|---|---|---|---|---|---|---|---|---|---|
| | | 2020 | | | | 2021 | | | |
| | | **BS** | **FS** | **GFS** | **MS** | **BS** | **FS** | **GFS** | **MS** |
| Harbin | CK | 101.1 e | 136.2 c | 125.2 c | 57.5 c | 105.6 e | 141.2 c | 126.6 e | 61.5 c |
| | SA1+PBF1 | 140.4 b | 154.7 b | 153.5 b | 69.2 b | 145.9 b | 161.7 b | 158.5 b | 75.0 b |
| | SA2+PBF2 | 158.5 a | 177.5 a | 165.4 a | 75.8 a | 164.0 a | 184.5 a | 170.4 a | 81.6 a |
| Daqing | CK | 105.8 d | 133.5 d | 108.5 d | 53.6 d | 109.8 d | 138.5 d | 110.5 e | 58.6 d |
| | SA1+PBF1 | 107.3 d | 140.8 c | 129.2 c | 58.5 c | 112.3 d | 146.8 c | 132.5 c | 64.3 c |
| | SA2+PBF2 | 123.7 c | 159.9 b | 153.8 b | 67.2 b | 128.7 c | 165.9 b | 157.1 b | 73.0 b |
| | L | * | * | * | * | * | * | * | * |
| | SA | * | * | * | * | * | * | * | * |
| Analysis | PBF | * | * | * | * | * | * | * | * |
| of | L × SA | ns | ns | ns | ns | ns | ns | ns | ns |
| variance | L × PBF | ns | ns | ns | ns | ns | ns | ns | ns |
| | SA × PBF | * | * | * | * | * | * | * | * |
| | L × SA × PBF | ns | ns | ns | ns | ns | ns | ns | ns |

At the research sites in Harbin and Daqing, the POD, CAT, and SOD activities of soybean leaves considerably improved from the BS to FS stages while sharply decreasing from the GFS to MS stages at levels of salicylic acid and pyraclostrobin. At various growth stages, the effects of SA1+PBF1 and SA2+PBF2 treatments on POD, CAT, and SOD in soybean leaves were significantly greater compared with HCK and DCK treatments. At the BS, FS, GFS, and MS stages of the research sites in Harbin and Daqing, SA1+PBF1 treatment did not significantly affect the POD, CAT, and SOD of soybean leaves.

Unlike the trend of changes in POD, CAT, and SOD activities, the MDA content in soybean leaves gradually improved from the BS stage to the MS stage (Figure 4). The MDA content decreased with the increase of salicylic acid and pyraclostrobin content. Under CK treatment, the MDA content in the MS phase of DCK treatment peaked. Under the control treatment, the MDA content in the study sites of Harbin and Daqing increased sharply, respectively. The maximum antioxidant enzyme activity was recorded throughout

the soybean growth season at the research sites in Harbin and Daqing, with the lowest MDA concentration observed in the SA1+PBF1 and SA2+PBF2 treatments.

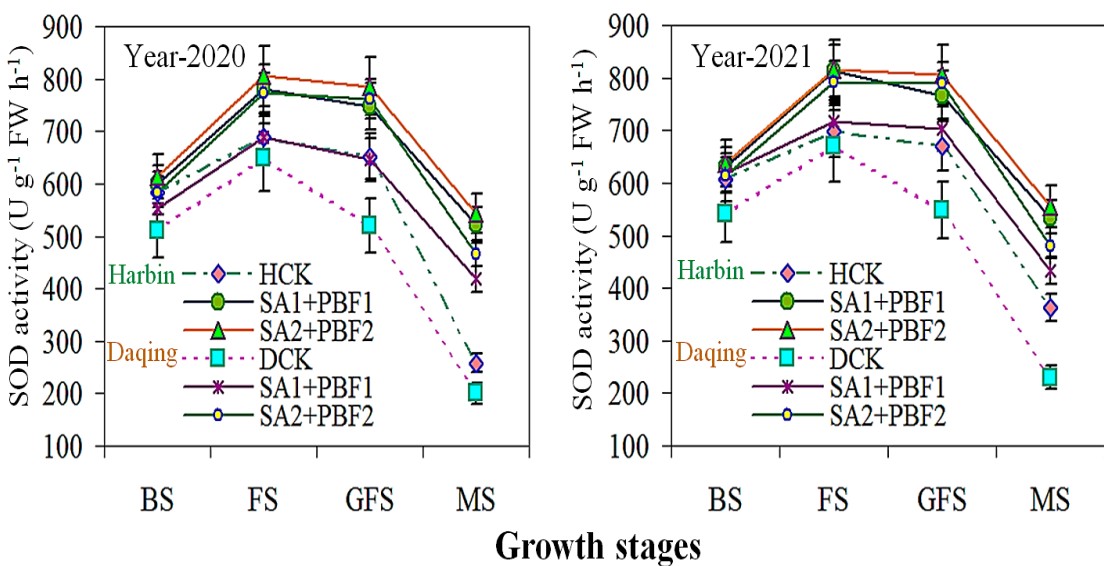

**Figure 3.** Effects of various treatments on superoxide dismutase activity of soybean during 2020–2021 growing seasons.

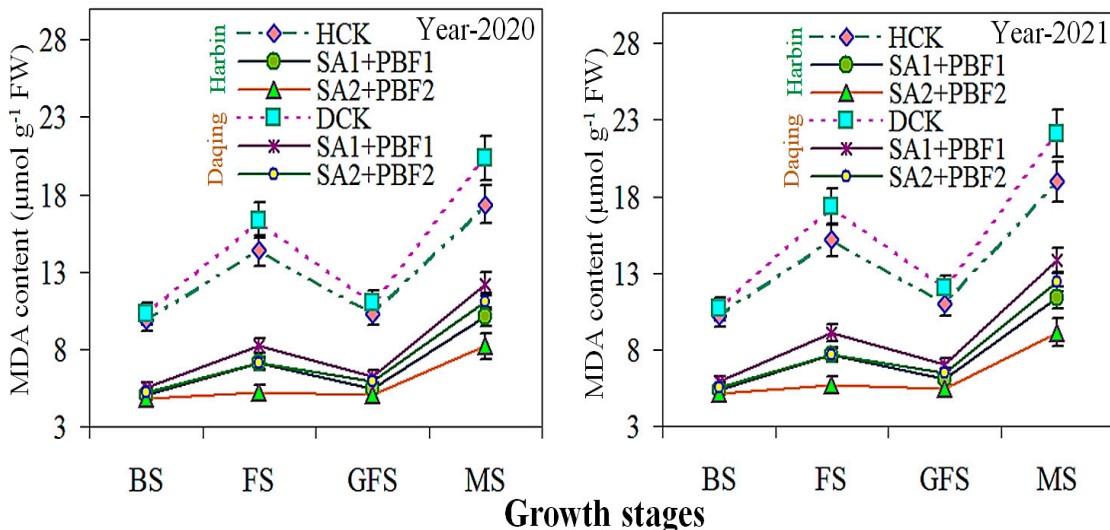

**Figure 4.** Effects of various treatments on malondialdehyde content of soybean during 2020–2021 growing seasons.

### 3.4. $H_2O_2$ and $O_2^-$ Contents

Under the HCK and DCK treatments, the $H_2O_2$ and $O_2^-$ of the soybean were significantly higher than that of SA1+PBF1 and SA2+PBF2 treatments throughout the soybean-growing season in both the Harbin and Daqing study locations (Figures 5 and 6). The $H_2O_2$ and $O_2^-$ of soybean slowly increased from the FS to the GFS and GFS to MS stages. Furthermore, under various salicylic acid and pyraclostrobin levels, the content of $H_2O_2$ and $O_2^-$ increased considerably from the GFS to MS stages. However, the average of two-year data at various growth stages of soybean indicated that in the location of Harbin, SA1+PBF1 and SA2+PBF2 treatments significantly decreased $H_2O_2$ by 47.3% and 28.8% compared with HCK treatment, while at the location of Daqing, SA1+PBF1 and SA2+PBF2 treatments significantly decreased $H_2O_2$ by 50.4% and 23.0% compared with DCK treatment. Meanwhile, the mean of $O_2^-$ content at various growth stages of soybean showed

that in the location of Harbin, SA1+PBF1 and SA2+PBF2 treatments significantly decreased $O_2^-$ by 53.2% and 35.6% compared with HCK treatment, while at the location of Daqing, SA1+PBF1 and SA2+PBF2 treatments significantly decreased $O_2^-$ by 57.3% and 30.8% compared with DCK treatment. At the BS, FS, GFS, and MS stages, the soybean leaves under the HCK and DCK were considerably higher in $H_2O_2$ and $O_2^-$ than in the SA1+PBF1 and SA2+PBF2 treatments. The $H_2O_2$ and $O_2^-$ of the SA1+PBF1 treatment were higher compared with SA2+PBF2 in both the Harbin and Daqing study locations.

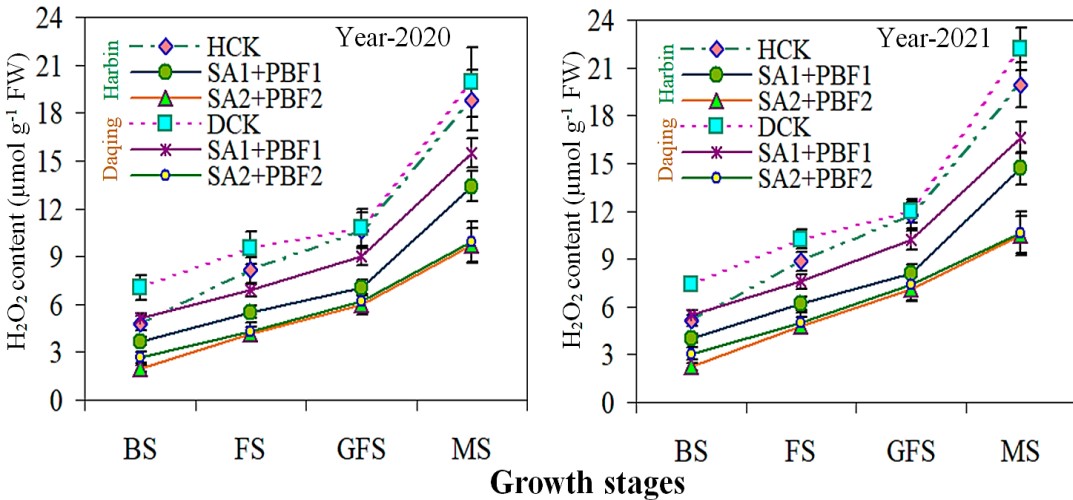

**Figure 5.** Effects of various treatments on hydrogen peroxide of soybean leaves during 2020–2021 growing seasons.

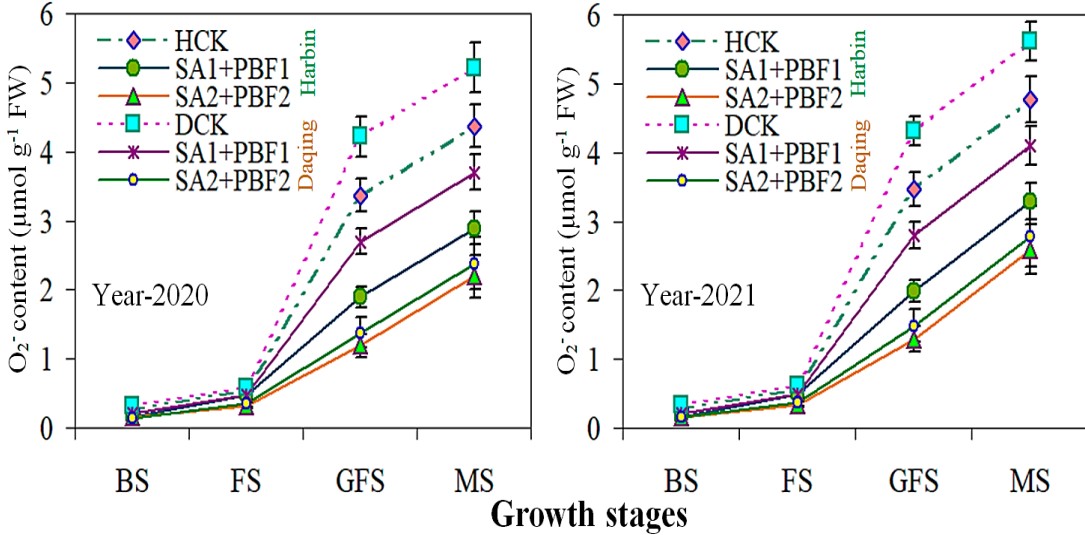

**Figure 6.** Effects of various treatments on superoxide radicals ($O_2^-$) of soybean during 2020–2021 growing seasons.

### 3.5. Yield and Yield Components

In both the Harbin and Daqing study locations, various salicylic acid and pyraclostrobin levels had considerable effects on grain plant$^{-1}$, 100-grain weight, WUE, and grain production (Table 6). In a two-year study, the results showed that compared with HCK and DCK treatments, SA2+PBF2 treatment improved SWS, Pn, and SP and reduced ET, thereby significantly improving the soybean yield under saline soil. Compared with HCK and DCK, the mean grain yield under the SA1+PBF1 and SA2+PBF2 treatments at both the Harbin and Daqing study locations improved by 29.2%, 32.1%, 28.9%, and 30.0%, while the average WUE of soybean was significantly increased by 37.1%, 43.2%, 35.5%, and

41.4%, respectively. The WUE indicated the relationship between water utilization and soybean production. Compared with all other treatments, SA2+PBF2 treatment considerably increased WUE at the Harbin study location.

**Table 6.** Soybean yield and yield components as affected by various treatments during 2020 and 2021.

| Locations | Treatments | Soybean Yield and Yield Components | | | | | | | | | |
| | | ET (mm) | | Grains Plant$^{-1}$ | | 100 Grain Weight (g) | | WUE (kg mm$^{-1}$ ha$^{-1}$) | | Grain Yield (t ha$^{-1}$) | |
| | | 2020 | 2021 | 2020 | 2021 | 2020 | 2021 | 2020 | 2021 | 2020 | 2021 |
|---|---|---|---|---|---|---|---|---|---|---|---|
| Harbin | CK | 366.5 b | 373.4 b | 128 c | 139 b | 21.60 b | 21.72 b | 6.68 c | 6.72 c | 2.45 e | 2.51 d |
| | SA1+PBF1 | 329.8 d | 360.4 c | 134 b | 140 b | 22.30 a | 22.70 b | 10.49 b | 9.85 b | 3.46 b | 3.55 b |
| | SA2+PBF2 | 296.0 f | 323.8 e | 142 a | 155 a | 23.30 a | 25.20 a | 12.16 a | 11.44 a | 3.60 a | 3.70 a |
| Daqing | CK | 378.2 a | 404.3 a | 83 g | 89 e | 18.20 c | 18.44 c | 5.15 d | 4.89 d | 1.95 f | 1.98 e |
| | SA1+PBF1 | 346.9 c | 363.1 c | 115 e | 117 d | 18.56 c | 21.18 b | 7.90 c | 7.66 c | 2.74 d | 2.78 c |
| | SA2+PBF2 | 308.9 e | 347.2 d | 122 d | 134 c | 20.90 b | 22.40 b | 9.03 b | 8.11 b | 2.79 c | 2.82 c |
| Analysis of variance | L | * | * | * | * | * | * | * | * | * | * |
| | SA | * | * | * | * | * | * | * | * | * | * |
| | PBF | * | * | * | * | * | * | * | * | * | * |
| | L × SA | ns | ns | ns | ns | ns | ns | ns | ns | ns | ns |
| | L × PBF | ns | ns | ns | ns | ns | ns | ns | ns | ns | ns |
| | SA × PBF | * | * | * | * | * | * | * | * | * | * |
| | L × SA × PBF | ns | ns | ns | ns | ns | ns | ns | ns | ns | ns |

## 4. Discussion

### 4.1. Photosynthesis Rate Responses to Different Salicylic Acid and Pyraclostrobin

With the increase of salinity treatment levels, the degree of growth reduction was higher. Several researchers reported on the growth and biomass decline of diverse plant species under soil salinity [45,46]. Ge and Zhang [47] proposed that growth delay was an important aspect of assessing the degree of damage caused by salinity, regardless of the type of plant species. In two study years, SWS varied with precipitation and different growth stages. From the SS to BS stages, the average SWS (0–2 m) of SA1+PBF1 and SA2+PBF2 treatments was significantly increased compared to HCK and DCK treatments in Harbin and Daqing. From the FS to GFS stages, SWS gradually increased. From the GFS stage to the MS stage, the SWS trend of each treatment improved compared to the FS stage. During the GFS to MS stages, the average SWS (0–200 cm) of SA2+PBF2 treatment in the Harbin region was significantly improved compared to the HCK treatment. Plastic film mulching can considerably increase SWS capacity and WUE [48,49]. Due to the higher SWS of the soil and the acceleration of soybean development, the soybean produces the maximum production [50]. Zhang et al. [51] also studied that after the application of SA, growth increased, and the ET rate decreased under salt stress. In the presented study, the soybean growth in the Daqing location had a high ET rate compared with the Harbin location under different salicylic acid and pyraclostrobin levels. The ET rate was significantly ($p < 0.05$) decreased at the location of Harbin with SA1+PBF1 and SA2+PBF2 treatments by 23.8% and 11.1% compared with HCK treatment, while at the location of Daqing, SA1+PBF1 and SA2+PBF2 treatments had significantly decreased the ET rate by 22.4% and 9.0% compared with DCK treatment. In addition, the effect of PBF on WUE confirms that the application of PBF can considerably increase WUE [52,53].

The toxic stress indicator can affect the content of Pn and SP in leaves [54]. The decrease in Pn can be attributed to the inhibition of Pn biosynthesis induced by saline [54]. The Pn and SP content of soybean was considerably improved with the increase of salicylic acid and pyraclostrobin levels. SP content is closely related to the photosynthetic rate, significantly increasing the photosynthetic rate of leaves. Due to the improvement of SWS, the SP content in soybean leaves was higher, which can maintain a higher net Pn rate. Under salt stress, PBF can improve the Pn rate [55]. Chlorophyll contents show a vital role in energy assimilation in plants, and their levels undergo significant changes under salt stress. During two years of the same treatment, the Pn and SP content in soybean leaves

was considerably higher from the BS to FS stages, while they decreased considerably from the FS to GFS stages. Many reports indicate that PGPB and SA are effective photosynthetic regulators as they have a positive impact on the structure of leaves and chloroplasts, as well as chlorophyll content [56]. Our research results also indicate that in the late growth stage of soybeans, Pn and SP content was significantly affected at each level by salicylic acid and pyraclostrobin, unlike in HCK and DCK treatments. In many studies, it has been observed that a salt-induced complete loss of some proteins increases the synthesis of new proteins, and it is believed that newly synthesized proteins play a crucial role in salt stress tolerance [57,58]. The surge in total soluble sugar content and the increase in salinity treatment levels may be due to the accumulation of starch and sugar under salt stress [59].

### 4.2. Antioxidative Enzyme Activities Responses to Different Salicylic Acid and Pyraclostrobin

In order to eliminate excessive ROS production in cells, plants have developed anti-oxidant enzymes such as CAT, POD, SOD, and ascorbic acid [54]. The SA2+PBF2 treatment had the highest POD, CAT, and SOD activities in soybean leaves during the FS stage, but there was no significant difference compared to the GFS stage. Afterward, the SOD, POD, and CAT in soybean leaves rapidly decreased during the MS stage. The $H_2O_2$ produced in cells under stress is removed through the action of POD, CAT, and SOD enzymes [60]. The increase in ascorbic acid levels is consistent with the higher levels of POD activity after the application of PBF and SA under salt stress. The slowing down of aging is also related to the decrease in lipid peroxidation caused by reduced oxidative stress [61,62]. In our experiment, it was observed that due to the application of PBF, the activity of antioxidant enzymes increased, which will confirm the latter hypothesis. Aging is considered a process related to reactive oxygen species [63]. Furthermore, there were no significant variances between the SA1+PBF1 treatments at all growth stages of soybeans at the research sites in Harbin and Daqing. At the research sites in Harbin and Daqing, the POD, CAT, and SOD activities of soybean considerably improved from the BS to FS stages, while sharply decreasing from the GFS to MS stages at levels of salicylic acid and pyraclostrobin. Water scarcity is associated with oxidative stress caused by increased ROS accumulation [64]. The oxidative damage can have a negative impact on Pn value [65]. However, crops have developed an anti-oxidant mechanism to decrease $H_2O_2$ and $O_2^-$ content [66]. The MDA decreases with the rise of salicylic acid and pyraclostrobin. The CAT and SOD increased, and reducing MDA, $H_2O_2$, and $O_2^-$ production may be a vital approach under salt stress for soybean to increase chlorophyll content [64].

Under the treatment of HCK and DCK, compared with the SA1+PBF1 and SA2+PBF2 treatments in the Harbin and Daqing research sites, the $H_2O_2$ and $O_2^-$ of soybean were considerably higher. The content of $H_2O_2$ and $O_2^-$ in soybean gradually increases from the FS to GFS and the GFS to MS stages. The accumulation of $H_2O_2$ and MDA under salt stress is the result of increased ROS production [67]. SOD, POD, and CAT are key enzymes in the reactive oxygen species scavenging system that can inhibit ROS accumulation. In fact, SOD catalyzes the dismutation of $H_2O_2$ and $O_2^-$, while the other three enzymes eliminate $H_2O_2$ and prevent its toxicity [66]. In addition, under different levels of salicylic acid and PBF, the $H_2O_2$ and $O_2^-$ significantly improved from the GFS to MS stages. At the research sites in Harbin and Daqing, compared to the SA2+PBF2 treatment, the SA1+PBF1 treatment had the highest levels of $H_2O_2$ and $O_2^-$. The salinity stress in this study resulted in a rise in $H_2O_2$ and MDA content. Various researchers have reported an increase in ROS production under salt stress [58,68]. The levels of MDA and $H_2O_2$ in plants treated with pyraclostrobin were lower than those affected by salt, indicating that PBF and SA significantly decreased lipid damage and prevented oxidative damage caused by salt stress [56].

### 4.3. Cultivation Systems Effect on Soil Water Storage and Soybean Production

The plastic film mulching considerably increased soybean yield, reduced ET, and effectively increased WUE [69]. Our results confirm that compared to HCK and DCK treatments, SA2+PBF2 treatment improved SWS, Pn, and SP and reduced ET, significantly

increasing soybean yield in saline–alkali soil. Compared with the HCK and DCK treatments, the grain yield of SA1+PBF1 and SA2+PBF2 treatments increased by 29.2%, 32.1%, 28.9%, and 30.0%, respectively, at the research sites in Harbin and Daqing, while the WUE of soybeans significantly improved by 37.1%, 43.2%, 35.5%, and 41.4%, respectively. However, the impact of RF systems on grain production is controversial. Previous research reported that RF systems considerably increase soybean production [63,70,71], while other studies have found that RF decreases the ET rate [72]. Our research results confirm that plastic film mulching can significantly improve water use efficiency, indicating a connection between water use and soybean production.

### 5. Conclusions

Salinity has become a potential risk to global agricultural productivity and food security. The research results indicate that in the Harbin region, SA2+PBF2 treatment reduced the ET rate, improved SWS during branching and grouting stages, and achieved a higher net photosynthesis rate. Moreover, this improvement is due to the reduction of oxidative damage and MDA in soybean at various growth stages. At different growth stages, the treatment of the Harbin soybean with SA2+PBF2 significantly increased the activity of SOD, POD, CAT, and SP, while the content of MDA, $H_2O_2$, and $O_2^-$ also decreased considerably. Under the SA2+PBF2 in Harbin, the scavenging ability of free $H_2O_2$ and $O_2^-$ was higher, and the activity of antioxidant enzymes was better. This was due to the lower level of lipid peroxidation, which efficiently secures the photosynthesis mechanism, considerably increasing WUE (46.3%) and grain yield (57.5%). These results indicate that the plastic mulch with SA2+PBF2 treatment significantly manages salt stress by improving SWS, anti-oxidant enzyme activities, photosynthesis, WUE, and soybean production to promote sustainable farming.

**Author Contributions:** H.R., J.W. and B.Z. conceived and designed the experiments. H.R., X.W., F.Z., K.Z., X.L., R.Y., C.Z., J.D. and J.Y. performed the experiments. H.R. and X.W. analyzed data and wrote the manuscript. All authors have read and agreed to the published version of the manuscript.

**Funding:** Project funded by Key R&D projects in Heilongjiang Province (Grant No. 2022ZX02B06); Key Laboratory of Soybean Mechanized Production, Ministry of Agriculture and Rural Affairs, China (Grant No. SMP202202); The National Key Research and Development Program of China (Grant No. 2022YFD1500505-1); Project of Natural Science Foundation of Heilongjiang Province (Grant No. ZD2020C009).

**Institutional Review Board Statement:** Not applicable.

**Informed Consent Statement:** Not applicable.

**Data Availability Statement:** Data will be available on personal request.

**Conflicts of Interest:** The authors declare no conflict of interest.

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
