# Peer review of "Salicylic Acid and Pyraclostrobin Can Mitigate Salinity Stress and Improve Anti-Oxidative Enzyme Activities, Photosynthesis, and Soybean Production under Saline–Alkali Regions"

_land, doi:10.3390/land12071319_

Round 1

Reviewer 1 Report

There is a lot of interesting data in the manuscript on soybean leaf antioxidant enzyme activities, photosynthesis and production in response to different levels of salicylic acid and pyraclostrobin and soil water availability. While the experimental design is balanced/appropriate and detailed time courses of various hormone / antioxidant enzyme activities are presented (every 3 days post silking / at 4 critical stages of development over 2 years) in addition to agronomic data (yield components). However, these below suggestions will help you to improve the quality of your research paper.

Title: The title clearly reflects the findings of the manuscript but it's not too long.

Abstract: In the abstract give some numbers reflecting your results. At the moment it's too general. Some abbreviations are not explained.

Key word: 6 key words are enough, so no need to reduce the key words.

Introduction: In the introduction you gave the importance to the China. How about the rest of the Word dry areas. Is this the common practice also there or something to suggest to other countries?

Last paragraph of your introduction the aims or objectives is not clear and well explain please rewrite your aims or objectives. If you add some latest literature's will be much more better if it available.

M&M: Material and methods are well described with valid statistical analysis and well explain procedure. Please add the procedure how you get soil moisture because soil moisture data is used to get soil water storage.

Results: Data are present on the 6 Figures and 7 Tables. Results are explained very well and clearly explain each tables and figures and much butter for general readers of LAND journal.

Discussion: The part of the manuscript, which should be re-structured with sub-title section, is discussion. I write three sub-titles please follow these sub-titles and re-structured your discussion part.

4.1. Photosynthesis rate responses to different Salicylic acid and pyraclostrobin

4.2. Antioxidative enzyme activities responses to different Salicylic acid and pyraclostrobin

4.3. Cultivation systems effect on soil water storage and soybean production

Author Response

Thank you to review our manuscript! You are kind and responsible reviewers, and the suggestions you have given are all valuable and very helpful for revising and improving our paper. We are very grateful to that. We have studied your comments carefully and have made correction; Many thanks to the Editor and the Reviewers for your time and thoughtful comments, many of which have been incorporated into the revised manuscript.

Below are our detailed responses (in BOLD type) to Editor and Reviewer’s comments, (the page and line numbers refer to our revised manuscript):

Reviewer #1: Author

General comments:

There is a lot of interesting data in the manuscript on soybean leaf antioxidant enzyme activities, photosynthesis and production in response to different levels of salicylic acid and pyraclostrobin and soil water availability. While the experimental design is balanced/appropriate and detailed time courses of various hormone / antioxidant enzyme activities are presented (every 3 days post silking / at 4 critical stages of development over 2 years) in addition to agronomic data (yield components). However, these below suggestions will help you to improve the quality of your research paper.

Response: respected reviewer, thank you very much for your encouragement and excellent suggestions.

Title: The title clearly reflects the findings of the manuscript but it's not too long.

Response: respected reviewer, thank you very much for your encouragement.

Abstract: In the abstract give some numbers reflecting your results. At the moment it's too general. Some abbreviations are not explained.

Response: Thank you for your advice, I rewrite and add some numbers which reflect my results, please see abstract in new revise version of my paper. I also explained the abbreviations of first time use in abstract.

Key word: 6 key words are enough, so no need to reduce the key words.
Response: Thank you for your advice.

Introduction: In the introduction you gave the importance to the China. How about the rest of the Word dry areas. Is this the common practice also there or something to suggest to other countries?
Response: Thank you for your advice, sir in introduction section I mention semi-arid regions of China so my research can be applied in the rest of the Word which are belong to semi-arid regions or dry land farming system.

Last paragraph of your introduction the aims or objectives is not clear and well explain please rewrite your aims or objectives. If you add some latest literature's will be much more better if it available.

Response: Thank you for your advice, sir according to your suggestion I rewrite my paper aims and objectives please see in new revise paper. I also add latest references in my new revise manuscript. 

M&M: Material and methods are well described with valid statistical analysis and well explain procedure. Please add the procedure how you get soil moisture because soil moisture data is used to get soil water storage.

Response: Thank you very much. Sir according to your advice I add the full length procedure of soil moisture, please see in a new revise manuscript.

Results: Data are present on the 6 Figures and 7 Tables. Results are explained very well and clearly explain each tables and figures and much butter for general readers of LAND journal.

Response: respected reviewer, thank you very much for your encouragement.

Discussion: The part of the manuscript, which should be re-structured with sub-title section, is discussion. I write three sub-titles please follow these sub-titles and re-structured your discussion part.

4.1. Photosynthesis rate responses to different Salicylic acid and pyraclostrobin

4.2. Antioxidative enzyme activities responses to different Salicylic acid and pyraclostrobin

4.3. Cultivation systems effect on soil water storage and soybean production

Response: Thank you for your advice, following your suggestion I re-structured the whole discussion with three sub-title section, please see in new revise manuscript.

Minor editing of English language required

Response: Thank you, following your suggestion, I corrected all grammar mistakes  according to your above suggestion, sir we have also sent our manuscript to a professional, native English-speaking Scientific Editor to improve the language and specifically to remove grammar mistakes.

Reviewer 2 Report

The manuscript has been well prepared, and the authors have conducted valuable research that can benefit soybean farmers in China. The study investigated the effects of salicylic acid and pyraclostrobin on photosynthesis, endogenous hormonal changes, and soybean production under salinity stress. The findings provide insights into how these substances can mitigate salinity stress, improve anti-oxidative enzyme activities, enhance photosynthesis, and increase soybean production in saline-alkali regions. Overall, the work is well reported and documented, and it is suitable for publication in the LAND Journal. However, there are several moderate flaws that need to be addressed:

Introduction:

Clearly state the research hypothesis/question and the objectives of the study. This will help readers understand the importance of your study and the differences from previous studies.

Experimental Design:

Quantify both salicylic acid (SA) and the given fungicide in the whole plant to understand the dynamics of these substances and their potential interactions.

Data Analysis:

Specify whether the normality of the data was tested, especially for enzyme or ROS quantification in leaves.

Typos and Errors:

Correct "rain-fid" to "rain-fed" and similar mistakes throughout the manuscript.

Ensure consistency in terminology, such as removing unnecessary occurrences of "Under."

Nitrogen Application:

Explain why nitrogen was not applied in a split application and discuss the potential risk of nutrient loss through leaching.

Soil Water Content:

Provide information on whether the soil water content meters were calibrated for the specific site.

Language and Style:

Edit the manuscript for grammar and clarity, and consider having it reviewed by a native English speaker to improve the language quality.

Please cite this paper. Soil extracellular enzyme activities under long-term fertilization management in the croplands of China: a meta-analysis

Conclusion:

Include a discussion of the impact of the study, particularly its contributions to grain yield under different treatments of salicylic acid and pyraclostrobin.

Moderate editing of English language required.

Author Response

Thank you to review our manuscript! You are kind and responsible reviewers, and the suggestions you have given are all valuable and very helpful for revising and improving our paper. We are very grateful to that. We have studied your comments carefully and have made correction; Many thanks to the Editor and the Reviewers for your time and thoughtful comments, many of which have been incorporated into the revised manuscript.

Below are our detailed responses (in BOLD type) to Editor and Reviewer’s comments, (the page and line numbers refer to our revised manuscript):

The manuscript has been well prepared, and the authors have conducted valuable research that can benefit soybean farmers in China. The study investigated the effects of salicylic acid and pyraclostrobin on photosynthesis, endogenous hormonal changes, and soybean production under salinity stress. The findings provide insights into how these substances can mitigate salinity stress, improve anti-oxidative enzyme activities, enhance photosynthesis, and increase soybean production in saline-alkali regions. Overall, the work is well reported and documented, and it is suitable for publication in the LAND Journal. However, there are several moderate flaws that need to be addressed:

 Response: respected reviewer, thank you very much for your encouragement and excellent suggestions.

Introduction:

Clearly state the research hypothesis/question and the objectives of the study. This will help readers understand the importance of your study and the differences from previous studies.

 Response: respected reviewer, thank you very much for your encouragement.

Experimental Design:

Quantify both salicylic acid (SA) and the given fungicide in the whole plant to understand the dynamics of these substances and their potential interactions.

 Response: respected reviewer, thank you very much for your encouragement.

Data Analysis:

Specify whether the normality of the data was tested, especially for enzyme or ROS quantification in leaves.

 Response: Thank you for your advice, yes sir the normality of data is analysis and tested and especially the enzyme or ROS quantification in soybean leaves.

Typos and Errors:

Correct "rain-fid" to "rain-fed" and similar mistakes throughout the manuscript.

Ensure consistency in terminology, such as removing unnecessary occurrences of "Under."

 Response: Thank you for your advice, sir according to your above suggestion the correct rain-fid into rain-fed, also I remove the unnecessary occurrences of “Under” word in my while article, sir please see in a new revise article.

Nitrogen Application:

Explain why nitrogen was not applied in a split application and discuss the potential risk of nutrient loss through leaching.

 Response: Thank you very much. Sir soybean is a nitrogen fixing plant, so soybean just need a starter dose of nitrogen, after that its fix the nitrogen with the help of root noodle, that is why we did not apply N in a split application, .

Soil Water Content:

Provide information on whether the soil water content meters were calibrated for the specific site.

 Response: Thank you very much. Sir according to your advice I add the full length procedure of soil moisture, please see in a new revise manuscript.

The soil water content was calculated at the seedling, branching, flowering, grain-filling stage, and maturity stage during 2020 and 2021 year. Moisture contents of the 0-200 cm soil layers at 20 cm intervals were recorded using a TDR meter (Time-Domain Reflectometry, Germany).

Language and Style:

Edit the manuscript for grammar and clarity, and consider having it reviewed by a native English speaker to improve the language quality.

 Response: Thank you, following your suggestion, I corrected all grammar mistakes  according to your above suggestion, sir we have also sent our manuscript to a professional, native English-speaking Scientific Editor to improve the language and specifically to remove grammar mistakes.

Please cite this paper. Soil extracellular enzyme activities under long-term fertilization management in the croplands of China: a meta-analysis

 Response: Thank you very much. Sir according to your advice I add the above article in my research article, sir really its help me a lot to improve my article, please see in a new revise manuscript.

Miao, F., Li, Y., Song, C., Sindhu, J., Guofeng, Y., Qingping, Z. 2019. Soil extracellular enzyme activities under long-term fertilization management in the croplands of China: a meta analysis. Nutr Cycl Agroecosyst, 114;125–138. https://doi.org/10.1007/s10705-019-09991-2

Conclusion:

Include a discussion of the impact of the study, particularly its contributions to grain yield under different treatments of salicylic acid and pyraclostrobin.

Response: Thank you, following your suggestion, I included a discussion of the impact of the study, particularly its contributions to grain yield under different treatments of salicylic acid and pyraclostrobin, please see in a new revise article.

Comments on the Quality of English Language

Moderate editing of English language required.

Response: Thank you, following your suggestion, I corrected all grammar mistakes  according to your above suggestion, sir we have also sent our manuscript to a professional, native English-speaking Scientific Editor to improve the language and specifically to remove grammar mistakes.
